# WHO ROUTES THE ROUTER: RETHINKING THE EVALUATION OF LLM ROUTING SYSTEMS

## ABSTRACT

The growing ecosystem of Large Language Models (LLMs) with diverse capabilities and costs has motivated the need for LLM routing systems that dynamically select the most appropriate model for each query. Evaluating these routing systems is important yet inherently challenging due to the complex interplay of multiple factors: the selection of representative input queries, the composition of the model pool, and the definition of comprehensive evaluation metrics for optimal routing decisions. Through extensive analysis of existing benchmarks, we identify critical limitations that may lead to incomplete results and/or misleading conclusions about router performance: (1) limited task diversity, (2) imbalanced model pools, and (3) oversimplified evaluation methodologies. To address these limitations, we propose a novel evaluation framework that incorporates diverse task distributions (33,337 queries across 68 categories), a balanced model pool of 85 models with complementary model strengths, and multi-faceted metrics that reflect real-world deployment scenarios. We implement this framework as an open-source benchmark, enabling researchers to rigorously assess routing strategies under realistic conditions. The code and dataset are shared anonymously at: `https://anonymous.4open.science/r/rethinking-routing-evaluation-DE30`

## 1 INTRODUCTION

Large Language Models (LLMs) are proliferating rapidly, resulting in a growing ecosystem of models with diverse parameter scales, capabilities, and computational costs (Varangot-Reille et al., 2025; Feng et al., 2025; Li, 2025). While this diversity offers rich options for model selection, it also raises a key question: *which model best achieves the desired performance while minimizing cost?* Today, a common practice is to use a single model to handle all requests. However, this approach faces inherent trade-offs between performance and cost, as no LLM is universally optimal. For example, massive models like GPT-4 excel at complex reasoning but are significantly more costly than smaller alternatives such as Mixtral-8×7B (Jiang et al., 2024) or Llama-3.1-8B (Grattafiori et al., 2024). Meanwhile, domain-specialized models often outperform general-purpose ones within their specific areas of expertise (Yang et al., 2024; Tu et al., 2024).

To address this, **LLM routing systems** (shown in Figure 1) have been proposed to dynamically select the most appropriate model for each query, matching query characteristics to model strengths while optimizing costs (Feng et al., 2025; Varangot-Reille et al., 2025; Li, 2025; Dekoninck et al., 2025; Stripelis et al., 2024; Wang et al., 2025; Diamond; NVIDIA). To design effective LLM routers, **rigorous evaluation becomes especially crucial** (Hu et al., 2024; Huang et al., 2025). Routing decisions directly impact user experience and system cost-efficiency, yet the complexity of query-model matching makes it difficult to reason about routing strategies analytically. Moreover, the rapid evolution of LLMs demands continual reassessment of these strategies. Without a standardized evaluation framework (Hu et al., 2024; Huang et al., 2025), it is challenging to compare different routing approaches, identify weaknesses, or confidently deploy systems where substantial costs and user satisfaction are at stake.

However, **evaluating LLM routers is inherently challenging**, as optimal routing decisions are context-dependent and shaped by specific priorities and constraints, such as cost, latency, and accuracy. Therefore, to build an effective evaluation framework that assesses router optimality across diverse scenarios, we must comprehensively consider the requirements from three core components of an LLM routing system: input queries/tasks, model candidates, and routing evaluation methodologies.

First, it requires carefully chosen input queries that reflect realistic usage patterns, encompassing a diverse range of tasks with varying levels of difficulty. Second, it requires a thoughtfully composed model pool that is sufficiently large and diverse—capturing variations in capabilities and costs—to enable meaningful routing decisions. Third, it requires effective evaluation methodologies that capture the multifaceted nature of routing performance, such as cost–performance trade-offs, latency constraints, and task accuracy.

Unfortunately, our extensive analysis of existing evaluation benchmarks, including RouterBench (Hu et al., 2024), LLM-Blender (Jiang et al., 2023), and EmbedLLM (Zhuang et al., 2024), reveals that current evaluation frameworks suffer from three fundamental shortcomings. (1) *Limited task diversity*: They often rely on artificially constructed tasks that fail to capture the complexity, diversity, and distribution of real-world queries. (2) *Imbalanced model pools*: They use imbalanced model sets where one model consistently outperforms others across all tasks, making routing decisions trivial. (3) *Oversimplified evaluation methodologies*: Many evaluations prioritize accuracy and aggregate metrics. While cost-aware analyses exist (e.g., ROUTERBENCH) and recent work proposes explicit cost–performance trade-offs, important aspects like routing-rate trade-offs, robustness under domain shift, and latency-awareness remain under-explored in a unified framework. Our work complements existing efforts by adding these facets.

These limitations underscore the urgent need for a more comprehensive and effective evaluation framework. In response, we introduce RouterBench+, a new, well-designed, and open-source benchmark that addresses these critical gaps and establishes a new standard for LLM routing evaluation. Our solution includes: (1) a **specialist-score-based task sampling method** that creates a diverse set of 33,337 queries across 68 categories, (2) a **similarity-aware greedy model pruning and extension strategy** that yields a balanced pool of 85 models with complementary strengths, and (3) a **comprehensive evaluation methodology** combining classification-based and routing-rate paradigms with explicit Out-of-Distribution (OOD) testing. Using our evaluation pipeline, researchers can rigorously assess routing strategies under realistic conditions and make informed decisions about model selection trade-offs. We summarize our contributions as follows:

- A systematic analysis of LLM routing evaluation requirements and key limitations in existing benchmarks, showing how current approaches overlook real-world challenges and can lead to misleading conclusions about router performance.
- A comprehensive evaluation methodology that includes three key aspects: diverse task distributions reflecting realistic query patterns; balanced model pools that avoid single-model dominance; and multi-faceted evaluation metrics that capture complex constraints and trade-offs.
- An open-source, extensible evaluation platform that implements our methodology, enabling rigorous routing evaluation under realistic conditions and helping toward designing optimal LLM routers.

## 2 PRELIMINARY: LLM ROUTING AND ITS IDEAL EVALUATION

### 2.1 FORMALIZATION OF LLM ROUTING AND OPTIMIZATION PROBLEM

An LLM routing system dynamically assigns a user query to the most suitable model from a pool of available LLMs under certain constraints. The system consists of:

**Input**: A user query $p \in \mathcal{P}$, represented by a query embedding $\mathbf{p} \in \mathbb{R}^d$.

**Model Pool**: A set of LLMs $\mathcal{M} = \{m_1, m_2, ..., m_n\}$ with per-query ground-truth quality $q_i(p)$ under the benchmark metric and per-query cost $c_i(p)$ (e.g., parameters, tokens, latency, or USD when available).

**Routing Function**: A routing function $R : \mathcal{P} \to \mathcal{M}$ that selects a model $m^* = R(p)$ for each input. This may be deterministic or probabilistic, outputting a distribution $s(p)$ over $\mathcal{M}$.

Routers are trained to estimate per-query quality $\hat{q}_i(p)$ (or confidence) for model $m_i$ given $\mathbf{p}$. We distinguish two objective views for clarity:

*(1) Per-query selection at a quality threshold.* Given a target quality threshold $T$, select the cheapest model meeting the threshold or, equivalently, maximize estimated quality:

$$m^* = \arg\min_{m_i \in \mathcal{M}} c_i(p) \text{ s.t. } \hat{q}_i(p) \geq T \quad \text{or} \quad m^* = \arg\max_{m_i \in \mathcal{M}} \hat{q}_i(p). \tag{1}$$

Figure 1: An illustration of LLM routing systems. An ideal LLM router should choose the model with highest expected performance under the specified constraints like costs.

*(2) Budgeted performance over a distribution of queries.* Under a budget $B$, optimize a (possibly stochastic) policy $s$ to maximize expected ground-truth quality while respecting expected cost:

$$\max_s \mathbb{E}_{p \sim \mathcal{P}} \Big[ \sum_{i=1}^n s_i(p)\, q_i(p) \Big] \quad \text{s.t.} \quad \mathbb{E}_{p \sim \mathcal{P}} \Big[ \sum_{i=1}^n s_i(p)\, c_i(p) \Big] \leq B. \tag{2}$$

In our experiments, we train using $\hat{q}_i(p)$ surrogates and *evaluate* with $q_i(p)$. We report deferral curves for (2) across budgets and routing-rate trade-off curves for fixed small/large pairs.

## 2.2 Ideal Router Evaluation

Figure 1 illustrates the architecture of an LLM routing system. An effective router should direct each query to the model with the highest expected performance while satisfying specified constraints like costs. To evaluate router performance and guide their design toward optimality, an effective and comprehensive evaluation framework is essential. However, designing such an evaluation framework presents inherent challenges. Unlike evaluating LLM performance—where each query can be assessed against a common ground truth—optimal routing strategies are highly context-specific. They depend on specific priorities and constraints, such as cost, latency, and accuracy. Even for the same query, the optimal routing decision may vary under different constraints.

To address these challenges, we return to the first principles by reconsidering what constitutes a "good router." We argue that ideal router evaluation should comprehensively assess routing strategies across diverse constraints and scenarios, providing clear differentiation between effective and suboptimal approaches. To this end, we distill three key requirements for a robust evaluation framework, which we examine in detail in the following sections.

- *Rich and realistic queries.* The task distribution should be diverse and representative of real-world usage, spanning various domains and difficulty levels. It should include both common and rare query patterns to evaluate router performance on familiar cases as well as unseen scenarios.
- *Diverse and balanced models.* The model pool should avoid single-model dominance, ensuring that each model has distinct strengths and weaknesses. It should include a mix of general-purpose and domain-specific models to ensure that routing decisions have a meaningful impact on performance.
- *Comprehensive evaluation metrics.* The evaluation framework should assess router effectiveness under varying constraints, capture trade-offs among performance, cost, and latency, and include OOD queries to evaluate robustness.

## 3 Related Work

**LLM Model Selection and Routing.** Intelligent LLM routers have emerged to route queries across diverse models to balance performance, cost, and latency (Feng et al., 2025; Varangot-Reille et al., 2025; Li, 2025; Yue et al., 2025; Zhang et al., 2025a). System-level cost-aware usage frameworks include FrugalGPT (Chen et al., 2023) and EcoAssistant (Zhang et al., 2023). Preference-data and contrastive approaches learn routing policies directly from feedback (Ong et al., 2025; Chen et al., 2024). Routing strategies can be categorized as predictive and non-predictive (Varangot-Reille et al., 2025; Hu et al., 2024). Predictive approaches include classification based on prompt features (Srivatsa et al., 2024), graph-based methods (GraphRouter (Feng et al., 2025)), dynamic routing (MixLLM (Wang et al., 2025)), and multi-armed bandit formulations (LLM Bandit (Li, 2025)). Non-predictive methods include cascading, while hybrid approaches like Cascade Routing (Dekoninck et al., 2025) combine routing flexibility with sequential processing. Frameworks like TensorOpera Router (Stripelis et al., 2024) further enhance multi-model inference efficiency. The proliferation of LLM routing methods has produced the requirement for effective router evaluation (Chen et al., 2024; Lu et al., 2024; Zhang et al., 2025b; Chuang et al., 2024).

**Benchmarks for Multi-LLM Systems.** Several benchmarks have been developed to evaluate routing strategies. RouterBench (Hu et al., 2024) provides a framework with inference outcomes across models and tasks (Dekoninck et al., 2025; Wang et al., 2025). EmbedLLM (Zhuang et al., 2024) introduces compact vector embeddings for efficient model selection. MixInstruct (Jiang et al., 2023) offers a mixture-of-instructions dataset with a two-stage ensembling approach. RouterEval (Huang et al., 2025) presents a large-scale benchmark with over 8,500 models and 200 million routing records. CARROT (Somerstep et al., 2025) proposes cost-advantage trade-off curves to quantify accuracy versus cost explicitly. Additional related work includes RouteLLM (Ong et al., 2025) and RouterDC (Chen et al., 2024). Shnitzer et al. (Shnitzer et al., 2023) discuss dataset construction for routing. These benchmarks are crucial for developing robust routing systems that enable cost-effective LLM deployment (Feng et al., 2025; Srivatsa et al., 2024; Varangot-Reille et al., 2025; Li, 2025).

Despite the growing body of work on LLM routing techniques and benchmarks, we identify a critical gap: **the evaluation methodology itself has not been systematically examined**. Even the most comprehensive and recently released benchmarks, such as RouterEval (Huang et al., 2025), primarily aggregate large volumes of data and models without addressing fundamental flaws in evaluation design. This paper fills that gap by critically analyzing current evaluation practices and providing concrete recommendations for improvement. In the following sections, we systematically examine the assumptions underlying current practices in query distribution, model selection, and evaluation metrics, highlighting how they can lead to misleading conclusions about router performance.

## 4 RETHINKING CURRENT EVALUATION PRACTICES

This section examines current LLM routing evaluations, beginning with an overview of our methodology. We then analyze the three core components of a routing system: tasks, models, and evaluation metrics. For each component, we (a) explain the underlying assumption or practice, (b) describe our experimental setup, including the dataset or benchmark used, and (c) present and discuss the results, highlighting what they reveal about the assumption.

### 4.1 EXPERIMENTAL SETUP

**Benchmark & Datasets.** We evaluate routing performance using three widely used benchmarks: EMBEDLLM (Zhuang et al., 2024), ROUTERBENCH (Hu et al., 2024), and MIXINSTRUCT (Jiang et al., 2023).

**Routing Methods.** The state-of-the-art LLM routing approaches can be broadly categorized into two groups: *clustering-based methods*, such as K-Means (Jitkrittum et al., 2025), K-NN (Hu et al., 2024); and *learning-based methods*, including MLP (Hu et al., 2024) and Collaborative Filtering (Matrix Factorization) (Zhuang et al., 2024). Additionally, we include two reference baselines to provide contextual performance benchmarks: a *Heuristic Router*, which routes all queries to the model with the highest average training accuracy within the allowed cost, and an *Oracle Router*, which serves as an upper bound by assuming access to ground-truth model performance at test time.

**Evaluation Metrics and Deferral Curve.** We evaluate routing performance on each benchmark using its corresponding evaluation metrics. For ROUTERBENCH (Hu et al., 2024) and EMBEDLLM (Zhuang et al., 2024), each LLM either answers a query correctly or not, producing a binary correctness label. For MIXINSTRUCT (Jiang et al., 2023), we follow prior work (Jitkrittum et al., 2025; Jiang et al., 2023) to adopt the exponentiated BARTScore for evaluation. Routing quality is visualized using a *deferral curve*, where the X-axis indicates the model cost budget, such as cost in dollars or parameter size; and the Y-axis represents routing quality, such as accuracy or exp(BARTScore). The deferral curve captures the trade-off between routing quality and resource usage, allowing comparison of different routing strategies under cost constraints.

For more details on the experimental setups, please refer to Appendix C–F.

### 4.2 TASKS: MORE DIVERSITY AND LESS REDUNDANCY

**Problem 1: Lack of Specialized Tasks.** Generally, LLM tasks can be categorized as *common-sense* or *domain-specific*. For instance, piqa (Bisk et al., 2019), a physical common-sense task, is handled well by general models (generalists), with an average accuracy of 78.03%. In contrast, the medical

Table 1: Comparison of routing performance before and after removing duplicate queries.

| Method | Avg. Acc. (%) ↑ | | Peak Acc. (%) ↑ | |
|---|---|---|---|---|
| | **Original** | **Reduced** | **Original** | **Reduced** |
| K-NN | **54.35** | 54.04 | **67.37** | 66.50 |
| Universal (KMeans) | **54.03** | 54.00 | **66.77** | 66.70 |
| MLP | 53.78 | **53.84** | 64.17 | **65.13** |
| Matrix Factorization | 50.48 | **50.90** | 60.07 | **60.87** |

domain's medmcqa (Pal et al., 2022) has a lower average of 41.73% with general models, while a domain-specific model (specialist) can achieve 69.8%.

To evaluate routing performance across different scenarios, the task set should be sufficiently diverse, including both common-sense and domain-specific tasks. However, current benchmarks such as ROUTERBENCH and EMBEDLLM are biased toward common-sense tasks. This bias could lead to a failure to evaluate routers' ability to handle domain-specific tasks, a critical class of queries that benefit a lot from model routing.

To quantify this imbalance, we propose a *specialist score* for each task: the average (across cost budgets) of the difference in accuracy between the best-performing domain-specific model and the best generalist (heuristic) model:

$$\text{specialist\_score}_{\text{task}} = \mathbb{E}_{b \in \mathcal{B}} \left[ \max_{m \in \mathcal{M}_{\text{non-gen}}^{(b)}} \text{ACC}_{m,t}^{(b)} - \text{ACC}_{\text{gen},t}^{(b)} \right],$$

where $\mathcal{B}$ represents cost budgets, $\mathcal{M}_{\text{non-gen}}^{(b)}$ excludes the general model, and ACC measures accuracy. This score captures how much specialists outperform generalists on specific tasks. We computed this score on both benchmarks; Figure 2 shows the results on EMBEDLLM, and the ROUTERBENCH counterpart is provided in Appendix G (Figure 9). Ideally, we expect a long-tail distribution—most tasks having moderate or negative scores, and a few showing high specialist scores—indicating that while general models suffice for many tasks, some benefit from specialization. However, we observed **only a limited number of specialist tasks across both benchmarks**.

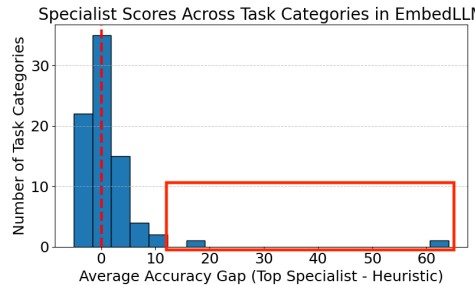

Figure 2: The specialist scores reveal that current datasets lack diverse, specialized tasks.

**Problem 2: Task Redundancy.** We have also found that current benchmarks suffer from significant task redundancy, which can lead to not only performance bias but also routers learning shortcuts. As detailed in Appendix G, many task categories exhibit high similarity in their average query embeddings. Even after removing duplicate categories from the training set, the router maintains strong performance. To investigate potential redundancy in the benchmarks, we moved from category-level analysis to query-level inspection, identifying semantically equivalent queries that appeared multiple times in the training set. We computed cosine similarity between normalized query embeddings. Queries $q_i$ and $q_j$ were considered duplicates if:

$$\text{sim}(q_i, q_j) = \frac{\langle \mathbf{e}_i, \mathbf{e}_j \rangle}{\|\mathbf{e}_i\| \cdot \|\mathbf{e}_j\|} \geq \delta, \quad \text{where} \quad \delta = 0.999$$

Our analysis revealed 1,346 duplicate groups (average size 2.7). Compounding this issue, 99.9% of these groups contained label disagreements across models—far exceeding the overall label mismatch rate of 37.7%. This suggests that semantically identical queries often received inconsistent labels. By removing such duplicates and retraining routers on the cleaned dataset, we observed improved performance for learning-based methods, as shown in Table 1. This confirms that **duplicate queries with conflicting labels can mislead routers** to learn from noise rather than meaningful patterns. We interpret these findings not as a formal proof, but as a strong signal that current evaluation protocols are brittle to such artifacts.

Table 2: Top-5 models by their average rank across tasks. Lower values indicate greater dominance.

| | EMBEDLLM | | | ROUTERBENCH | |
|---|---|---|---|---|---|
| Rank | Model ID | Avg. Rank ($\downarrow$) | Rank | Model ID | Avg. Rank ($\downarrow$) |
| 1 | 50 | 6.43 | 1 | 5 | 1.36 |
| 2 | 83 | 9.88 | 2 | 10 | 3.20 |
| 3 | 42 | 10.03 | 3 | 4 | 3.78 |
| 4 | 49 | 10.95 | 4 | 9 | 3.88 |
| 5 | 5 | 11.24 | 5 | 3 | 5.39 |

Table 3: Selected tasks for pseudo specialist models.

| Task | Prompt % | Mean Acc. (%) | Best Model Acc. (%) | Pseudo Model Acc. (%) |
|---|---|---|---|---|
| Social Reasoning | 5.42 | 33.76 | 36.22 | **65.00** |
| Logical Reasoning | 1.82 | 28.28 | 45.93 | **70.00** |
| Graduatel-Level Reasoning | 3.23 | 22.44 | 33.51 | **60.00** |

**Insights.** Current benchmarks overestimate the value of large but non-diverse training sets; in reality, much of the routing signal is concentrated in a smaller, more representative subset of tasks. To build more effective routing benchmarks, we should improve task diversity—especially by including more domain-specific tasks—and reduce redundancy, particularly tasks with inconsistent labels.

### 4.3 MODELS: MORE SPECIALISTS AND LESS DOMINANCE

**Problem 1: Model Dominance.** A meaningful model pool should ensure that each LLM contributes unique strengths—some serving as generalists, others as specialists. This diversity is essential to the routing task: matching each input to the most capable model. If a single model dominates across all tasks, routing becomes redundant.

To quantify dominance, we compute each model's *average rank* across task categories. Table 2 shows that EMBEDLLM (112 models) has multiple competitive models, whereas ROUTERBENCH (11 models) has a single generalist with average rank 1.36.

In a well-constructed benchmark or a more realistic routing scenario, some tasks (e.g., symbolic math, medicine, or historical reasoning) should require domain-specific expertise that only specialist models can provide. This mirrors the real-world objective of a router finding a small yet expert model for a given task. However, in current benchmarks, strong generalist models often fill this role, even for tasks they were not explicitly designed for. This reduces the routing objective to simply identifying the best generalist, undermining the value of fine-grained model selection.

**Effective Expert Model Extension.** We propose augmenting the model pool with *pseudo-specialist models* as a diagnostic stress test. Unlike real models, these are not meant for deployment but serve as a methodological probe to simulate a scenario where distinct specialists exist. In standard benchmarks, single-model dominance often compresses the performance gap between simple heuristics and sophisticated routers. By surgically introducing pseudo-specialists, we create a controlled environment to rigorously test a router's *adaptability*—its ability to detect and leverage a specialist when one is actually available. We select three target tasks that are challenging (with low mean accuracy), non-dominated (having a modest best–mean gap), and have non-negligible representation in the benchmark.

We inject three pseudo-specialist models to break single-model dominance and diagnose routers' ability to select specialists. For a chosen target task $t$ meeting criteria (Table 3), we set the pseudo-model accuracy close to BestAcc$(t) + 25\%$ and set average accuracy for other tasks.

We further define the *agreement score* as the average percentage of queries for which a router selects the same model as the heuristic router. This metric reflects how closely a learned router mimics static generalist selection. A lower score indicates more diverse, task-specific choices, suggesting less reliance on the generalist strategy. As shown in Table 4, overall agreement with the heuristic router drops slightly

Table 4: Changes in router agreement with the top-1 generalist model after adding pseudo specialist models. Negative values indicate decreased reliance on the dominant model.

| Task | K-NN | KMeans | MF | MLP |
|---|---|---|---|---|
| Overall | -0.84 | -2.40 | -0.64 | **-8.40** |
| logiqa | **-20.55** | **-31.03** | -2.81 | **-17.48** |
| social_iqa | -2.69 | 0.00 | +0.29 | **-7.92** |
| gpqa | -1.59 | **-13.15** | +1.90 | **-9.75** |

across all methods. However, on the tasks targeted by the pseudo models, the reduction is significantly more pronounced.

**Problem 2: Model Redundancy.** We also observe redundancy in the model pool, which adds little value to training or evaluating router performance. We quantify model-level similarity using a Jaccard-style score based on shared correct predictions as detailed in Appendix H. We apply this strategy to the `EmbedLLM` benchmark, reducing the model pool from 112 to 82 (a 27% reduction). The experimental result (in Appendix H) shows that routing performance across methods remains comparable to the full model pool. This shows that removing redundant models does not degrade routing quality and that meaningful routing decisions can still be made with a leaner model pool.

**Insights.** Effective routing evaluation depends on a model pool with *meaningful* diversity, both in capability and specialization. Rather than including many models with overlapping strengths, the pool should consist of models with distinct specialties. A simple yet effective way to enhance current model pools is to introduce pseudo-specialist models that simulate task-specific expertise, encouraging routers to move beyond generic selection and make more nuanced, task-aware decisions.

## 4.4 EVALUATION PARADIGMS: COMPREHENSIVE MEASUREMENTS

**Problems**. Current evaluation paradigms still have two gaps: (1) *Incomplete cost awareness*: Beyond aggregate cost–accuracy curves, evaluations rarely measure explicit routing-rate trade-offs (how accuracy evolves with the fraction of queries deferred to a more expensive model). (2) *Lack of OOD evaluation*: Frameworks seldom test router performance on OOD inputs, an essential aspect for ensuring robustness in real-world deployments.

**Multi-Faceted Evaluation**: We argue that model routing evaluation should be multi-faceted, which should employ metrics that capture both performance quality and resource efficiency:

- *Cost-aware evaluation*: It should implement evaluation scenarios that explicitly consider cost constraints and encourage efficient model selection.
- *OOD testing framework*: It should develop a systematic approach to evaluate router performance on OOD scenarios, ensuring robustness in real-world deployment.

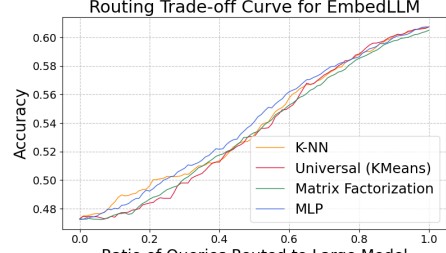

Figure 3: Binary routing evaluation paradigm showing performance trade-offs, between Llama-2 7B and Llama-2 70B

**Routing Tradeoff Evaluation:** To complement traditional cost-accuracy deferral curves, we introduce a binary routing evaluation paradigm to assess how effectively a router balances between a strong generalist (large model) and a lightweight alternative (small model). This connects to cost-advantage curves (Somerstep et al., 2025) but specializes to routing-rate control for a fixed model pair. For each router, we vary the fraction of queries routed to the small model based on the router's confidence and measure the resulting accuracy. This produces a continuous trade-off curve between routing accuracy and reliance on expensive models, as shown in Figure 3.

**Paradigm Distinction.** The deferral curve asks: "What accuracy at a given budget across the full pool?" The binary routing trade-off asks: "For a fixed small/large pair, how does accuracy change as we reduce reliance on the large model?" We use the former for overall budgeted performance, and the latter to diagnose cost-efficiency under single dominant model scenarios.

**Per-Query Cost and CoT.** While we primarily use parameter count as a proxy for cost/latency, our framework supports per-query metrics (tokens, wall-clock latency) and dynamic strategies (e.g., CoT), which we discuss in Appendix F.

**OOD Testing Framework:** In real-world deployment, routers are likely to encounter out-of-distribution (OOD) queries—inputs from domains or tasks not represented during training. While benchmark designers should strive to include diverse tasks to improve generalization, OOD inputs are inevitable given the open-ended

Table 5: OOD Performance change on math-related categories in EMBEDLLM when these categories are excluded from training.

| Category | K-NN Δ | KMeans Δ | MF Δ | MLP Δ |
|---|---|---|---|---|
| mathqa | -9.29 | -16.88 | -6.33 | -14.34 |
| asdiv | -58.59 | -69.19 | -40.40 | -57.07 |
| gsm8k | -14.28 | -14.29 | -29.47 | -35.72 |

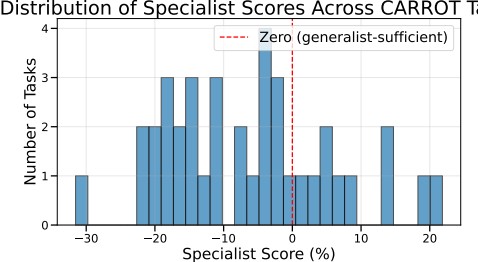

Figure 4: Distribution of specialist scores across CARROT tasks. Most tasks cluster at or below zero (generalist-sufficient), with only a small right tail of clearly specialist-demanding tasks.

| Rank | Model | Avg Rank |
|------|-------|----------|
| 1 | Qwen2-72B-Instruct | 2.97 |
| 2 | Llama-3.1-70B-Instruct | 3.80 |
| 3 | Qwen2.5-72B-Instruct | 4.93 |
| 4 | WizardLM-2-8x22B | 5.70 |
| 5 | QwQ-32B-Preview | 6.09 |

Figure 5: Top-5 most dominant models in CARROT dataset.

nature of user interactions with LLMs. Thus, evaluating router robustness in such scenarios is crucial. We design an OOD evaluation setup by holding out a subset of queries (e.g., Math tasks in EMBEDLLM) from training and evaluating performance on them separately. Table 5 illustrates an example split; we can see that different methods have different abilities for the OOD task.

**Insights.** Current benchmarks inadequately assess the ability of router in realistic scenarios. The OOD performance degradation (Table 5) reveals the brittleness of routers with novel queries, highlighting the need for better generalization testing. Additionally, the binary routing paradigm (Figure 3) shows that routing algorithms have distinct efficiency-performance trade-offs, requiring evaluation beyond single-point metrics.

### 4.5 ADDITIONAL ANALYSIS ON CARROT

**Task diversity and lack of specialized tasks.** We first apply the specialist-score analysis from Section 4.2 to the CARROT (sprout) tasks (Somerstep et al., 2025), using Qwen/Qwen2.5-72B-Instruct as the generalist model. Figure 4 shows the distribution of specialist scores across 38 tasks (excluding `ifeval`, which reports a scalar correctness). The distribution is strongly concentrated at and below zero: 28 tasks (73.7%) have *negative* specialist scores, meaning that the generalist outperforms all available specialists on these tasks, and 3 tasks (7.9%) have small positive scores in $[0, 0.05)$, indicating that specialists offer only marginal gains. Only 7 tasks (18.4%), mostly math-heavy or structured reasoning benchmarks (e.g., `math_prealgebra_hard`, `math_algebra_hard`, and `math_num_theory_hard`), are clearly specialist-demanding with scores $\geq 0.05$. Thus, while CARROT does contain some specialist-oriented tasks, the majority of its queries are generalist-sufficient, providing only moderate coverage of the specialization regime that routing methods are designed to exploit (Figure 4).

**Duplicated and near-identical Queries.** Our analysis on CARROT applies this same procedure in Section 4.2 over the 16,852 training queries. We identify 954 duplicate groups at a cosine-similarity threshold of $\delta = 0.999$, covering 2,293 queries in total (13.6% of the training set) with an average group size of 2.40 and a maximum size of 33. Strikingly, *every* duplicate group (100%) exhibits label disagreements, almost always both within a single model and across models—for example, identical math problems appearing in both `leaderboard_math_algebra_hard` and `leaderboard_mmlu_pro`, or the same GPQA question replicated across the `main`, `extended`, and `diamond` splits but assigned inconsistent correctness labels. These findings indicate that CARROT contains a non-trivial amount of query duplication with noisy labels, which can encourage routers to memorize artifacts rather than learn robust decision rules, and motivates deduplication when using CARROT for training and evaluating routing methods.

**Model Dominance.** We also examine model dominance on CARROT by computing each model's average rank across the 38 tasks (Section 4.3). Table 5 reports the top-5 models. The most dominant model, Qwen/Qwen2-72B-Instruct, attains an average rank of 2.97 across all tasks, followed by Meta-Llama-3.1-70B-Instruct (3.80) and Qwen2.5-72B-Instruct (4.93). Unlike ROUTERBENCH, where a single model has an average rank close to 1, CARROT exhibits a *moderately competitive*

landscape: several large models remain close in performance, and no single generalist uniformly wins every task. This reduces the risk that routing collapses to always choosing one dominant model and instead preserves headroom for routers to exploit nuanced trade-offs among strong but differently skilled LLMs.

# 5 REMASTERED EVALUATION PIPELINE

Building on our analysis of current evaluation limitations and the ideal characteristics of router evaluation, we present a comprehensive framework for assessing LLM routing systems. While developing a "perfect" evaluation pipeline presents challenges comparable to designing an "ideal" LLM router itself, we provide a framework that addresses the key shortcomings identified in our experimental analysis.

## 5.1 BENCHMARK DESIGN

Our evaluation framework is built upon core principles that directly address the limitations identified in our experimental analysis, as shown in Figure 6.

**Diverse task distributions:** Drawing from our findings on data representation issues, we incorporate **tasks with varying levels of difficulty, domain coverage, and redundancy** ❶ to reflect real-world scenarios where task distributions are rarely static or uniform. This addresses the limitations identified in our analysis of current benchmarks that assume representative and static task distributions. To achieve this, we subsample tasks and queries from EMBEDLLM using the proposed *specialist score*, highlighting tasks where non-generalist models provide additional values. This results in a task pool that emphasizes both broad coverage and the need for routing.

**Balanced model pool:** To reduce single-model dominance observed in some benchmarks, we **curate model pools that increase meaningful specialization and diversity** ❷. This design choice enables rigorous evaluation of fine-grained routing decisions. We apply the greedy model pruning strategy discussed in Sec 4.3 to eliminate redundant models, using a similarity-aware scoring function balancing accuracy and uniqueness. This reduces 30 models from the model pool. Additionally, we introduce three *pseudo specialist models* targeting challenging tasks (Table 3) to diversify the routing model pool.

For reproducibility and extensibility, we release the complete model list in the repository (EmbedLLM/data/model_order.csv) and mirror it in Appendix. The tooling supports modular updates to the pool as LLM capabilities evolve.

**Multi-faceted evaluation metrics:** Responding to our findings about oversimplified evaluation approaches, we combine **both classification-based and routing-rate paradigms** ❸ to provide a comprehensive assessment of router performance under different constraints. This moves beyond binary evaluation approaches that fail to capture the complexity of real-world routing scenarios, incorporating critical factors like cost-performance trade-offs, latency constraints, and reliability under varying workloads. We integrate cost-constrained evaluation with routing-rate analysis to provide researchers with multiple perspectives on router performance. Furthermore, our metrics specifically **account for OOD performance** ❹, ensuring that routers are evaluated on their ability to generalize to novel scenarios. We include dedicated OOD testing phases that assess router performance on novel task types and difficulty levels, providing insights into real-world deployment readiness.

The final dataset has 85 models, 68 categories, and 33,337 queries, in total of 3 million datapoints.

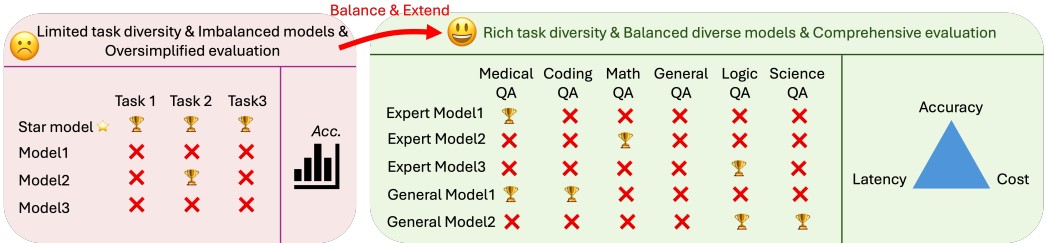

Figure 6: The improvements of our proposed benchmark.

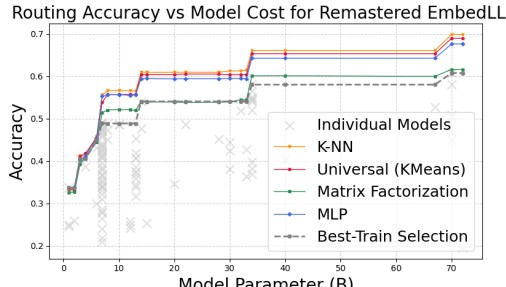

(a) Deferral curve on Remastered Benchmark

| Method | Area ↑ | Peak Acc. (%) ↑ |
|---|---|---|
| K-NN | 0.567 | 69.83 |
| Universal (KMeans) | 0.560 | 68.93 |
| MLP | 0.554 | 67.60 |
| Matrix Factorization | 0.515 | 61.60 |
| Heuristic | 0.507 | 60.73 |

(b) Area and peak accuracy of routing methods

Figure 7: Routing performance on our Remastered Benchmark. For deferral curve, the X-axis represents the model parameter constraint (cost proxy), and the Y-axis shows the achieved accuracy under that constraint.

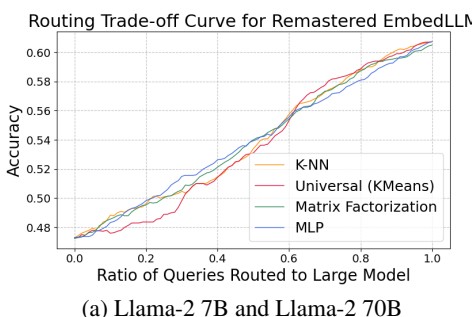

(a) Llama-2 7B and Llama-2 70B

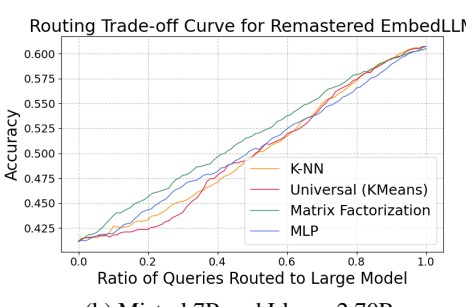

(b) Mistral 7B and Llama-2 70B

Figure 8: Binary routing evaluation on Remastered Benchmark shows performance trade-offs.

## 5.2 EXPERIMENT RESULTS

We evaluate routing methods on our remastered benchmark, with results shown in Figure 7. K-NN achieves the highest performance with an area under the deferral curve of 0.567 and peak accuracy of 69.83%. Figure 7(a) illustrates the performance-deferral trade-offs, demonstrating that our dataset has successfully mitigated the single model dominance problem. Figure 8 shows the binary routing paradigm results, which reveal distinct efficiency-performance patterns across different model combinations. One reason for this performance ranking is that the test prompts form tight neighborhoods in the embedding space. K-NN leverages local similarity to aggregate per-neighborhood model performance. Learning-based approaches (e.g., MLP, matrix factorization) smooth over the input space and may underuse sharp local signals when clusters are tight and heterogeneous across tasks. For the full results and additional evidence showcasing the improvements of our benchmark, please refer to the Appendix.

## 6 CONCLUSION

In this work, we have conducted a comprehensive analysis of LLM routing evaluation practices and identified critical limitations in current benchmarks. Through extensive experimentation, we demonstrated that existing evaluation frameworks often fail to capture the true complexity of routing decisions, leading to potentially misleading conclusions about router performance. Our findings reveal fundamental issues in task distribution representation, model pool composition, and evaluation metrics that significantly impact the validity and effectiveness of routing system evaluation. To address these limitations, we proposed a novel evaluation framework that incorporates diverse task distributions, balanced model pools, and multi-faceted metrics, providing researchers and practitioners with a more robust tool for assessing routing strategies. This work not only advances our understanding of what constitutes effective LLM routing but also establishes a foundation for more rigorous evaluation practices in this rapidly evolving field.

ETHICS STATEMENT

All authors have read and will adhere to the ICLR Code of Ethics (`https://iclr.cc/public/CodeOfEthics`). This work uses publicly available benchmarks (EMBEDLLM, ROUTERBENCH, and MIXINSTRUCT) as cited; no human subjects research, personal data, or sensitive information was collected. We report results and release an anonymous repository to enable scrutiny and responsible use. We are not proposing deployment guidance for safety-critical settings; our analysis focuses on evaluation methodology. The authors are not aware of conflicts of interest, and all experiments comply with dataset licenses and legal requirements.

REPRODUCIBILITY STATEMENT

We provide an anonymous repository with code, configuration files, and scripts to reproduce data subsampling, model-pool pruning/extension, metric computation, and figures; see the link in the abstract. Implementation details for encoders, benchmarks, routing methods, and evaluation metrics are described in Sections 5.1 and Appendices C, D, E, and F. The repository includes the full model list (e.g., `EmbedLLM/data/model_order.csv`), dataset splits, and seeds to facilitate end-to-end replication.

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

## A  THE USE OF LLMS

We used a large language model only for spelling and grammar correction of the manuscript text. The LLM was not involved in research ideation, experimental design, analysis, or substantive writing beyond copy-editing. All content and claims were authored and verified by the authors, who take full responsibility for the paper. The LLM is not an author.

## B  DISCUSSION AND LIMITATIONS

Our benchmark is a diagnostic tool designed to reveal routing behaviors under controlled conditions. First, we use pseudo-specialist models to simulate clear specialist advantages when real specialist models are scarce; we study sensitivity to pseudo accuracy settings and plan replacement with real specialists. Second, beyond parameter counts as a reproducible proxy, our framework supports per-query costs (tokens, measured latency, USD) and dynamic strategies (e.g., CoT) via standardized logging hooks. Third, results depend on the model pool and task mix; we release the full pool, enable modular updates, and report bootstrap confidence intervals for all AUC and routing-rate comparisons.

Despite these contributions, several fundamental challenges remain. The absence of a universal ground truth for optimal routing decisions means that what constitutes "optimal" depends heavily on specific deployment contexts. Specifically, we rely on reproducible metrics (binary labels, BARTScore) which may not fully capture open-ended conversation quality, a trade-off we accept for standardized benchmarking. The rapid evolution of model pools makes it challenging to maintain stable evaluation benchmarks, while the difficulty in capturing the full spectrum of real-world query patterns and task distributions remains a persistent issue. Additionally, the inherent trade-offs between different evaluation metrics (e.g., cost vs. performance) present ongoing challenges for both researchers and practitioners. While these limitations highlight the dynamic nature of LLM routing evaluation, they also emphasize the need for continued research in this area as the field evolves alongside the rapid development of LLM.

## C  DETAILS ABOUT TEXT ENCODER

Text encoder is a critical component of LLM routers, which transforms input prompts into embeddings used for routing decisions. To ensure faithful and fair comparison, we follow prior work (Zhuang et al., 2024; Hu et al., 2024) and adopt consistent encoder choices per benchmark: we use `all-MiniLM-L12-v2` (Sentence-Transformers, 2021a) for ROUTERBENCH and MIXINSTRUCT, and `all-mpnet-base-v2` (Sentence-Transformers, 2021b) for EMBEDLLM.

While we standardize on these encoders to isolate routing logic, we acknowledge that embedding choice is a significant factor influencing router performance. Future work can explore how different embeddings (e.g., instruction-tuned or domain-specific encoders) affect routing accuracy.

## D  DETAILS ABOUT BENCHMARKS

Table 6 summarizes the statistics of used benchmarks. EmbedLLM provides the largest number of models, while RouterBench provides a realistic cost setting. **MixInstruct** focuses on open-domain user prompts, using soft metrics like BARTScore to evaluate output quality.

Table 6: Comparison of benchmark datasets for LLM routing evaluation.

| Benchmark | # Models | # Queries | # Categories | Metric | Cost Info |
|---|---|---|---|---|---|
| **EmbedLLM (Zhuang et al., 2024)** | 112 | 35,673 | 80 | Binary (0/1) | param size (B) |
| **RouterBench (Hu et al., 2024)** | 11 | 36,497 | 86 | Binary (0/1) | USD per 1k queries |
| **MixInstruct (Jiang et al., 2023)** | 12 | 110,000 | 5 (Open-domain) | exp(BARTScore) | param size (B) |

# E    DETAILS ABOUT ROUTING METHODS

The state-of-the-art LLM routing approaches fall into two primary categories: *clustering-based* and *learning-based*. We also include two *reference baselines* to contextualize performance.

- **K-Means** (Jitkrittum et al., 2025): This method clusters training queries into $K$ clusters based on their embeddings. Given a test query $q$, the router finds the closest cluster $C_k$ and selects the model $m^*$ that performs best on average within that cluster:

$$m^* = \arg \max_{m_i \in \mathcal{M}} \left[ \frac{1}{|C_k|} \sum_{l \in C_k} \text{metric}(m_i, l) \right]$$

  where $C_k$ is the set of training prompts in the cluster of $q$, and metric denotes either a binary correctness label or $\exp(\text{BARTScore})$.

- **K-NN** (Hu et al., 2024): Instead of relying on cluster centroids, this method finds the $K$ nearest neighbors of the query $q$ in the training set (based on embedding distance) and routes to the model with the highest average score on those neighbors.

- **MLP** (Hu et al., 2024): For each LLM $m_i$, a separate MLP is trained to predict the performance score for query $q$:

$$P_i(x) = f(W_n \cdot \sigma(\dots \sigma(W_1 \cdot x + b_1) \dots) + b_n)$$

  where $x$ is the query embedding, $\sigma$ denotes the activation function, and $f$ is the final output layer. The model $m^*$ with the highest predicted score $P_i(q)$ is selected.

- **Collaborative Filtering (Matrix Factorization)** (Zhuang et al., 2024): This method treats the model routing task as a matrix completion problem. Given a binary matrix $Y \in \{0,1\}^{M \times Q}$ representing whether model $m_i$ correctly answered query $q_j$, it learns latent embeddings for models and queries by factorizing $Y$ as:

$$Y_{ij} \approx u_i^\top v_j$$

  where $u_i \in \mathbb{R}^d$ is the latent embedding for model $m_i$ and $v_j \in \mathbb{R}^d$ for query $q_j$. At inference time, the router computes $v_q$ (e.g., via a linear projection from query embedding) and selects the model with the highest predicted score:

$$m^* = \arg \max_{m_i \in \mathcal{M}} u_i^\top v_q$$

- **Heuristic Router**: This baseline selects the best-performing model from the training set for each cost budget. At each test time cost step, it routes all queries to the model that achieved the highest average training accuracy within the allowed cost:

$$m^* = \arg \max_{m_i \in \mathcal{M}, \, \text{cost}(m_i) \leq c} \text{TrainAcc}(m_i)$$

- **Oracle Router**: This upper-bound baseline assumes access to the ground truth performance of all models at test time. For each query, it routes to the best model among those allowed by the cost constraint:

$$m^* = \arg \max_{m_i \in \mathcal{M}, \, \text{cost}(m_i) \leq c} \text{metric}(m_i, q)$$

It represents the best possible routing performance under the given budget.

# F    DETAILS ABOUT EVALUATION METRICS AND DEFERREL CURVE

**Evaluation Metric.** We evaluate routing performance using metrics aligned with each benchmark's design. For RouterBench (Hu et al., 2024) and EmbedLLM (Zhuang et al., 2024), the correctness label is binary—each LLM either answers a query correctly or not. For MixInstruct (Jiang et al., 2023), we adopt the exponentiated BARTScore, following prior work (Jitkrittum et al., 2025; Jiang et al., 2023). While MixInstruct was originally intended to benchmark ensemble generation quality from outputs of multiple LLMs, recent works have adapted it for routing by assigning scores to individual LLM responses based on similarity to GPT-4. However, this introduces a dependency on GPT-4 as a reference model, which we will discuss further in Section 4.4.

**Deferral Curve.** Routing quality is visualized using a *deferral curve*, where the x-axis corresponds to the model cost budget and the y-axis reflects routing quality (accuracy or exp(BARTScore)). The cost budget represents the maximum cost (e.g., in dollars) a router can spend per query. However, because actual API pricing varies and is not always available, prior work (Jitkrittum et al., 2025) approximates cost using the number of model parameters—a practical proxy that correlates with both latency and financial cost for EmbedLLM (Zhuang et al., 2024) and MixInstruct (Jiang et al., 2023). This deferral curve captures the trade-off between routing quality and resource usage, allowing comparison of different routing strategies under cost constraints.

**Statistical Testing.** To compare curves, we report the area under the deferral curve (AUC) with 95% bootstrap confidence intervals over queries. Unless otherwise specified, we use paired bootstrap resampling (10,000 samples) of test queries and recompute method AUCs per sample; differences are deemed significant when the 95% CI of the pairwise AUC difference excludes zero. As a sanity check, we include a random-routing baseline (uniform over models within budget). We additionally apply this procedure to routing-rate trade-off curves by integrating accuracy over deferral rates.

**Per-Query Cost and Latency.** Beyond parameter counts, our framework can evaluate per-query costs using: (i) token-level accounting (prompt and completion tokens), (ii) measured wall-clock latency, and (iii) USD cost when API pricing is available. Concretely, we support logging a per-query tuple $(c^{\text{params}}, c^{\text{tokens}}, c^{\text{latency}}, c^{\$})$ and computing deferral curves and routing-rate trade-offs under each cost. This enables analyses aligned with CARROT-style cost-advantage curves while remaining reproducible across open-source and API models.

## G  SUPPLEMENTARY RESULT FOR TASK DIVERSITY

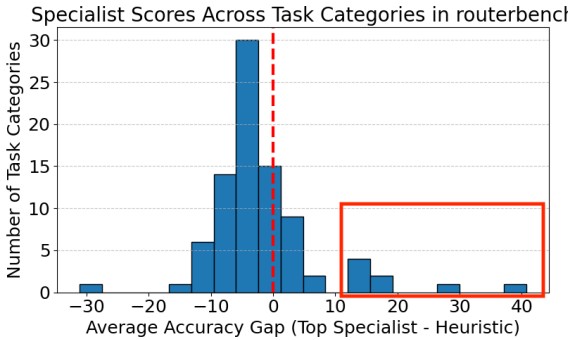

Figure 9: Specialist scores on ROUTERBENCH reveal limited specialized tasks.

Here we provide more results and discussions on **Task Imbalance and Redundancy Problem** in Section 4.2. As shown in Figure 10, several task categories exhibit high similarity in their average query embeddings. For instance, GPQA-like categories cluster tightly in the embedding space, suggesting that they may not offer distinct routing challenges.

In our experiments, we found that even after removing duplicate categories from the training set— those identified as redundant in the heatmap—the router still performs strongly. Figure 11 shows that this holds true even under OOD evaluation, where the dropped categories are tested at inference time. This suggests that current benchmarks may overestimate the value of large or diverse-looking training sets when, in reality, much of the routing signal is concentrated in a smaller, more representative subset of tasks. We also empirically assess the redundancy within categories, where we progressively dropped a portion of training data within each category and retrained the router.

## H  SUPPLEMENTARY RESULT FOR MODEL DIVERSITY

Here we provide more results and discussions **Model Redundancy Problem** in Section 4.3. We observed redundancy in the model pool, as evidenced by overlapping performance points (Individual

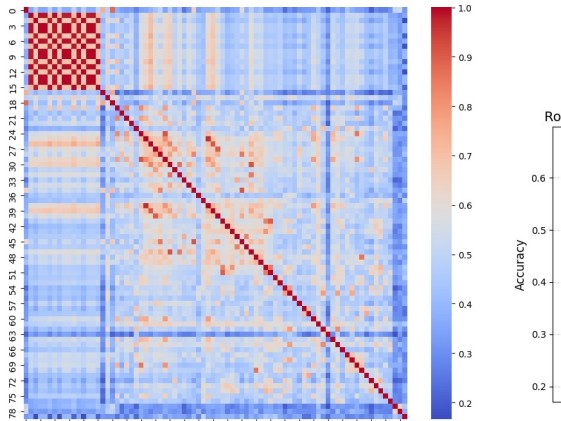

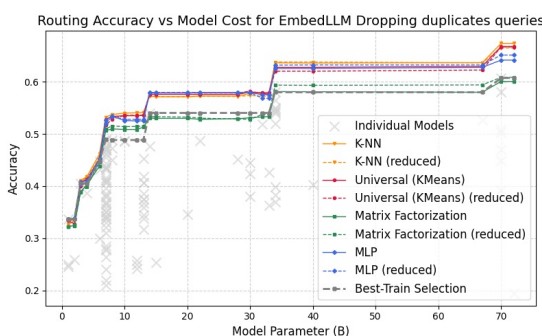

Figure 10: Category similarity heatmap based on average query embeddings. Redundancy is visible across GPQA-like categories (Upper-Left).

Figure 11: Routing accuracy when removing duplicate categories (e.g., GPQA variants). Performance is preserved even under OOD evaluation.

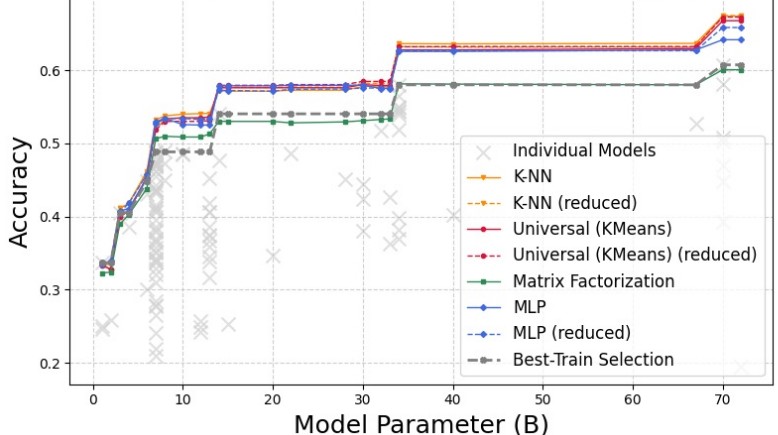

Figure 12: Performance comparison after reducing the model pool by 30 models. This shows that routers can maintain routing effectiveness across different cost budgets.

Models' grey crossings) across cost settings in Figure 12. Such redundancy adds little value for training or evaluating router performance. We quantify model-level similarity using a Jaccard-style score based on shared correct predictions:

$$\text{sim}(m_i, m_j) = \frac{|\{q \mid m_i(q) = 1 \wedge m_j(q) = 1\}|}{|\{q \mid m_i(q) = 1 \vee m_j(q) = 1\}|}$$

where $m_i(q)$ denotes whether model $m_i$ answered query $q$ correctly. This metric captures functional overlap across the entire benchmark.

To validate this, we propose a greedy pruning strategy to reduce model redundancy while preserving routing effectiveness. At each step, we compute a score for each model based on:

$$\text{score}(m_i) = \lambda \cdot \text{Accuracy}(m_i) - (1 - \lambda) \cdot \text{AvgSim}(m_i)$$

where $\text{AvgSim}(m_i)$ is the average Jaccard similarity of model $m_i$ to all other models (based on overlapping correct predictions), and $\lambda$ balances performance versus uniqueness. The model with the lowest score is removed, and the process repeats until a target number of models remains.

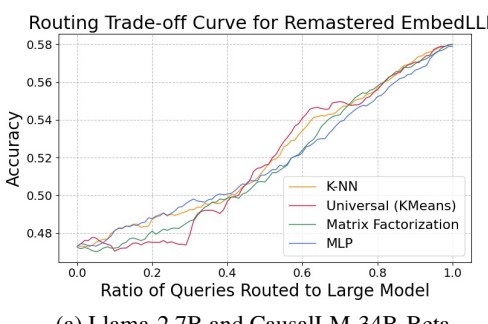 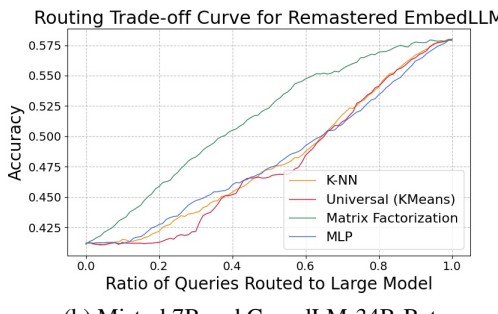

(a) Llama-2 7B and CausalLM-34B-Beta      (b) Mistral 7B and CausalLM-34B-Beta

Figure 13: Binary routing evaluation paradigm showing performance trade-offs.

We apply this strategy to the `EmbedLLM` benchmark, reducing the model pool from 112 to 82 (a 27% reduction). As shown in Figure 12, routing performance across methods remains comparable to the full model pool. This demonstrates that removing redundant models does not degrade routing quality and that meaningful routing decisions can still be made with a leaner model pool.

# I  SUPPLEMENTARY RESULT FOR EVALUATION METHODOLOGY

## I.1  BINARY ROUTING EVALUATION DETAILS

To complement the traditional cost-accuracy deferral curves, we introduced a binary routing evaluation paradigm in Section 4.4 to assess how effectively a router balances between a strong generalist (large model) and a lightweight alternative (small model). Here, we provide additional details about the evaluation setup and key observations.

We fix the large model to be `CausalLM-34B-Beta`, given its similar superior performance across a wide range of general-purpose tasks, comparable to that of 70B-sized models. For small models, we consider two widely used options: `Mistral-7b-v0.1` and `LLaMA-2-7b-chat-hf`. These models represent different trade-offs in model families and capability, making them ideal candidates for evaluating routing flexibility.

In this setting, each routing method ranks the queries by its confidence score for the small model and routes a varying fraction of queries accordingly, as in Figure 13. The remaining queries are deferred to the large model. This produces a continuous accuracy curve as a function of the fraction of queries routed to the large model.

Across both small model settings, we observe that learned routers generally follow a linear trade-off curve, indicating that they lack precise mechanisms to identify which queries can be reliably handled by the small model. Notably, clustering-based methods perform sub-linearly at lower deferral ratios, suggesting they often misclassify harder queries as easy ones and route them to the small models. This reinforces the need for more fine-grained routing strategies that can better distinguish between simple and complex inputs. Surprisingly, Matrix Factorization performed extremely well on classifying between Mistral-7B and CausalLM-34B-Beta, suggesting the potential of learning-based methods in certain model pair settings.

## I.2  OOD ROUTING EVALUATION DETAILS

We evaluate the robustness of routing methods under out-of-distribution (OOD) scenarios by training and evaluating routers on different domains. We consider two distinct OOD settings: (1) excluding all math-related queries (e.g., `mathqa`, `asdiv`, `gsm8k`), and (2) excluding all medical-related queries (e.g., `medmcqa`, `mmlu_clinical_knowledge`). These categories are chosen for their semantic distinctiveness and task specificity, providing strong settings to evaluate how well routers generalize to unseen topics.

As shown in Table 7, all routing methods suffer performance degradation in OOD settings, with the most significant drops occurring on `asdiv` and `gsm8k`. MLP-based routers tend to experience the

Table 7: OOD Performance change on selected categories in EMBEDLLM when these categories are excluded from training.

| Category | K-NN △ | KMeans △ | MF △ | MLP △ |
|---|---|---|---|---|
| mathqa | -9.29 | -16.88 | -6.33 | -14.34 |
| asdiv | -58.59 | -69.19 | -40.40 | -57.07 |
| gsm8k | -14.28 | -14.29 | -29.47 | -35.72 |
| medmcqa | -11.58 | -7.91 | -6.78 | -9.89 |
| mmlu_clinical_knowledge | 0.00 | +7.41 | -14.82 | -3.70 |
| **Average** | -18.75 | -20.17 | -19.56 | -24.14 |

steepest accuracy declines overall, while matrix factorization (MF) demonstrates greater robustness, particularly on math-related tasks.

These results highlight that existing routing strategies are brittle when deployed in domains unseen during training, reinforcing the need for more semantically aware or domain-adaptive routing mechanisms.

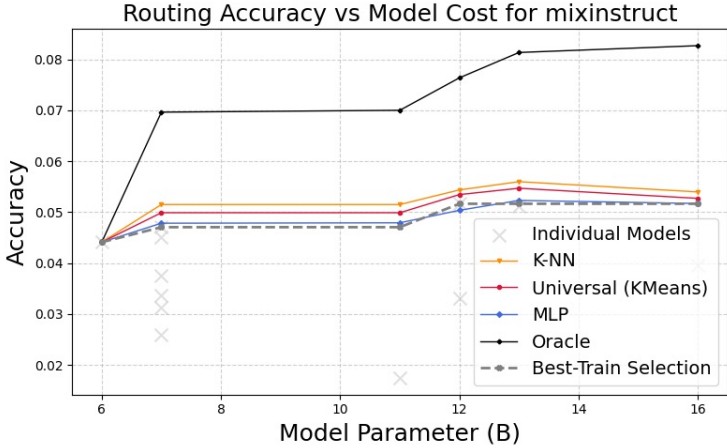

Figure 14: Routing performance on MIX-INSTRUCT.

## J   SUPPLEMENTARY RESULT ON MIX-INSTRUCT

In Figure 14, we present the routing results in deferral curve on MIX-INSTRUCT dataset. While the same baseline routers are evaluated, we do not consider MIX-INSTRUCT as our primary benchmark due to several limitations:

- **Limited Evaluation Metrics:** MIX-INSTRUCT uses BARTScore to measure the similarity between a model's output and a reference response generated by GPT-4. This approach conflates model quality with similarity to GPT-4, making it less suitable for evaluating true routing performance. It favors models that mimic GPT-4's phrasing—even when other models might generate more informative or appropriate responses—thus undermining the purpose of routing for capability-based model selection.
- **Limited Task Diversity:** The benchmark contains only five tasks, all of which fall under casual or instruction-following dialog. These tasks do not capture the breadth of real-world user queries, particularly in domains requiring specialized knowledge (e.g., science, math, law), thereby limiting the opportunity for routing to leverage model specialization.
- **Restricted Model Pool:** MIX-INSTRUCT covers about 10 models—comparable to Router-Bench—restricting the expressiveness of routing policies. In contrast, EMBEDLLM benchmark includes over 100 models with diverse strengths while having some issues we listed in Section 4, offered a more realistic and rigorous setting for evaluating routing capabilities.

