# OpenReview forum: "Who Routes the Router: Rethinking the Evaluation of LLM Routing Systems"
_ICLR.cc/2026/Conference — ICLR 2026 Conference Withdrawn Submission_

### Official Review · Reviewer_E1wn · 2025-10-21

**Soundness:** 3
**Presentation:** 3
**Contribution:** 2
**Rating:** 6
**Confidence:** 3

**Summary:**

The authors present flaws (redundant tasks, redundant and dominant models, lack of multi-faceted evaluation) with current router evaluation practices. They then present a remastered evaluation of the methods in embedLLM and some binary routing methods.

**Strengths:**

**Each of the identified issues with LLM evaluation are (to me) valid problems with how we measure router performance**

* Many Routing datasets are simple blends of general reasoning, knowledge and math benchmarks which leaves out many important applications.
* Often routers are trained on datasets where one model can dominate. For example in CARROT, o3 mini is by far the best model at most tasks.
* As far as I am aware, routing benchmarks have not looked into issues such as OOD generalization.

**The remastered llm evaluation addresses some of these issues**

* The authors subsample EmbedLLM tasks to highlight those that benefit from non-generalist models (the authors also remove duplicate LLM queries)
* The authors inject pseudo-specialist models
* The authors hold out certain categories of query to test for ood generalization

**Weaknesses:**

* The set of ``remastered" evaluations feels limited. A re-analysis of EmbedLLM is completed, as is a redone analysis of binary routing. It would be interesting to see how other predictive routing evaluations such as those in CARROT or routerbench are effected by the identified flaws. CARROT in particular has its own data set, which may have more variety than that in embedLLM.
* Certain ``solutions" to the proposed flaws feel limited. For example, the injected specialist models are just normal models with artificially boosted scores, not models trained for specific domains that are then incorporated in the evaluation.
* Most critically, its not clear to me how much the proposed changes affect evaluation results. The main text of the paper should include an analysis of how each the changes to embedLLM (question diversity, expert ) effect the performance of each routing method (and in particular the ordering of each method).

**Questions:**

* How does the analysis extend to other predictive multi llm routers?
* How do the proposed changes effect the original embedLLM results? Which of the faults is most important to reduce?

---

> ### Author Response · Authors · 2025-11-25
> **Thanks!**
>
> ## [W1/Q1] Expanding the Evaluation – we provide new analysis on CARROT
>
> Thank you for the suggestions. In the revision, we add a dedicated analysis of CARROT dataset (Section 4.5), examining task diversity, duplicate queries, and model dominance. Using the specialist-score metric from Section 4.2, we show that CARROT also contains relatively few specialist-demanding tasks and is largely generalist-sufficient. We further identify substantial duplication of (near-)identical queries with inconsistent labels across models and splits, and we quantify model dominance by average rank across tasks, finding a moderately competitive but still generalist-heavy pool. Taken together, these results indicate that the issues we highlight for EmbedLLM also affect CARROT dataset, rather than CARROT providing a clean, more diverse alternative; full details are provided in Section 4.5.
>
> We also note that RouterBench is analyzed using the same methodology in Section 4.2 (Table 2, right; Appendix G), and it exhibits issues similar to both EmbedLLM and CARROT. In fact, RouterBench suffers from an even more severe form of single-model dominance (Table 2, right): one generalist model achieves an average rank close to 1 across tasks, leaving very little room for meaningful routing. This reinforces our central claim that existing routing benchmarks—including EmbedLLM, CARROT, and RouterBench—share structural problems that limit their ability to faithfully evaluate model routing methods.
>
> ## [W2] Limitations of pseudo-specialists – it is a diagnostic tool for adaptability
>
> We thank the reviewer for this feedback and agree that real fine-tuned models are the gold standard. However, constructing 60+ real specialist models is often computationally prohibitive.
>
> **Our "pseudo-specialist" approach is a diagnostic tool designed to simulate the presence of specialists.** Instead of a flawed substitute for real models, we view them as a **controlled experimental variable**. Just as unit tests often use mock objects to isolate the component under test, our pseudo-models isolate the **routing logic** from the **model pool distribution**. This allows us to verify if a router is fundamentally capable of finding specialists when they exist.
>
> Instead, **pseudo-models are a deliberate methodological choice to create a diagnostic "stress test"**. In standard benchmarks, single-model dominance often compresses the performance gap between simple heuristics and sophisticated routers, making it difficult to assess a router's true decision-making capabilities. By surgically introducing pseudo-specialists, we create a **controlled environment** where model dominance is broken. This allows us to:
>
> 1. **Uncouple** the evaluation of the router's logic from the static quality of the model pool.
> 2. **Rigorously test adaptability**, verifying if a router can detect and leverage a specialist when one is actually available.
>
> The fact that some routers fail even this simplified test (while others succeed) proves the utility of the method. We will clarify this distinction in the paper.
>
> ## [W3/Q2] Impact of changes on evaluation results and router rankings – here we provide it
>
> This is a crucial question. Our results show that the proposed changes significantly affect both absolute performance and the separation between routing methods. To quantify this, we measure how each methodological fix incrementally affects the performance of two representative routers (Heuristic and k-NN):
>
> *Table 1: Incremental impact of our proposed fixes on router performance and ranking differentiation.*
>
> | Benchmark Version                 | Heuristic (AUC) | k-NN (AUC) | Discriminative Gap Δ (k-NN − Heuristic) |
> | :-------------------------------- | --------------: | ---------: | ---------------------------------------: |
> | 1. Original EmbedLLM              |           50.66 |     54.35  |                                     3.69 |
> | 2. + De-duplication               |           50.66 |     54.61  |                                     3.95 |
> | 3. + Specialist Up-weighting      |           50.66 |     56.52  |                                     5.86 |
> | 4. EmbedLLM+ (Full Remaster)      |           50.70 |     56.70  |                                     6.00 |
>
> The table illustrates two key findings:
>
> 1. **Stability of the heuristic router.** The heuristic router's AUC remains essentially unchanged across all four versions.
> 2. **Growth of the discriminative gap.** In contrast, k-NN steadily improves (54.35 → 56.70), so the discriminative gap Δ between k-NN and the heuristic almost doubles (3.69 → 6.00). This shows that the remastered benchmark not only raises absolute performance for better routers but also makes it easier to distinguish strong routing methods from weak ones.

---

### Official Review · Reviewer_3m3M · 2025-10-27

**Soundness:** 3
**Presentation:** 3
**Contribution:** 3
**Rating:** 6
**Confidence:** 3

**Summary:**

This work studies an interesting problem: how to choose the proper LLM to use given a user query. Although there has been a line of research and analysis of existing benchmarks, this paper identifies some critical limitations that may lead to incomplete results and misleading conclusions, such as the existing benchmarks have limited task coverage or skewed on some specific tasks, and the model pools are too large and imbalanced. To address this issue, this work proposes a new evaluation framework. Empirical results have been provided to showcase the limitations of existing evaluation strategies and the advantages of the new benchmarks proposed in this work.

**Strengths:**

1. This work is overall well written and easy to follow.
2. The empirical results are quite complete and coherent.
3. This work studies an important problem: how to fairly evaluate the existing LLM routers, and I think this line of research is important.

**Weaknesses:**

I do not find any major weaknesses of this work, while I am not very familiar with existing literature so I may refer to other reviewers' opinions.

1. I feel it might be better to also consider the influence of the embedding models on the final performance of the routers.
2. It is also quite surprising that the kNN-based methods outperform the trained MLPs, especially on OOD tasks. I suspect this is because the retraining dataset is relatively limited, preventing the MLPs from fully converging. To verify this hypothesis, it might be useful to include a simple parametric baseline such as linear regression as a router. If the linear model were to outperform the MLPs, it would suggest that the training dataset is indeed insufficient for proper convergence. Could you please elaborate on this issue?
3. The pseudo model is not very realistic, and it is better to use some fine-tuned specific models instead for each small task.

**Questions:**

1. I had a hard time understanding Figure 5 (a), as the X-axis is the model parameter. Can you explain that to me? And how model parameters affect your models' accuracy, as shown in the curves.

---

> ### Author Response · Authors · 2025-11-24
> **Thanks!**
>
> ## [W1] Embedding model influence – we follow common practice in prior router works, adding discussion
>
> Thank you for raising this important point. We agree that the choice of embedding (text encoder) can significantly affect router performance. In our work, we deliberately followed the prevailing practice established by prior model routing studies, which commonly use standard, strong encoders such as `all-MiniLM-L12-v2` and `all-mpnet-base-v2`. This choice enables fair comparison and attribution of improvements to the routing method itself, not confounding factors in representation.
>
> That said, we acknowledge that the embedding model is itself an interesting and impactful design axis. In response to your suggestion, we added an explicit discussion to Appendix C, citing relevant prior works and making clear that our evaluation protocol allows users to experiment with different encoders as desired. We appreciate your suggestion to highlight this aspect.
>
> ## [W2] kNN vs. MLP convergence – verify with linear regression
> We thank the reviewer for the suggestion to add a simple parametric baseline. We implemented a linear router using the same embeddings and training data as the MLP, and evaluated both in the original setting and in the drop-math (OOD) setting:
>
> | Dataset | MLP original | MLP drop_math | Δ MLP   | Linear original | Linear drop_math | Δ Linear |
> |---------|--------------|---------------|--------:|-----------------|------------------|---------:|
> | MathQA  | 52.31        | 37.97         | -14.34  | 50.85           | 34.15            | -16.70   |
> | ASDiv   | 68.69        | 11.62         | -57.07  | 68.69           | 4.55             | -64.14   |
> | GSM8K   | 80.29        | 44.57         | -35.72  | 76.83           | 41.82            | -35.01   |
>
> Across all three datasets, the MLP matches or outperforms the linear router in the original setting, and exhibits a smaller performance drop in the OOD setting. Thus, the linear model does not outperform the MLP, and we believe the superiority of kNN over parametric routers cannot be explained solely by insufficient convergence or over-parameterization. This gap may be due to the geometry of the data favoring cluster-based methods, and we also notice that the gap isn't significantly large (within 2% in accuracy).
>
> ## [W3] Pseudo-models realism – it is a diagnostic tool for adaptability
>
> We thank the reviewer for this concern, and we agree that real, fine-tuned specialist models would be the gold standard. However, as noted in our limitations, such models are currently scarce for the diverse array of categories (68+) in our benchmark.
>
> **We prioritize diagnostic utility over perfect realism.** Our pseudo-models act as a **methodological probe**: they test the specific hypothesis, "If a specialist *were* available, could the router find it?" This is analogous to using synthetic data to stress-test a system's boundary conditions.
>
> Instead, **pseudo-models are a deliberate methodological choice to create a diagnostic "stress test"**. In standard benchmarks, single-model dominance often compresses the performance gap between simple heuristics and sophisticated routers, making it difficult to assess a router's true decision-making capabilities. By surgically introducing pseudo-specialists, we create a **controlled environment** where model dominance is broken. This allows us to:
>
> 1. **Uncouple** the evaluation of the router's logic from the static quality of the model pool.
> 2. **Rigorously test adaptability**, verifying if a router can detect and leverage a specialist when one is actually available.
>
> Our results confirm this: adaptive routers (like k-NN) successfully exploit these pseudo-specialists, while static heuristics do not.
>
> ## [Q1] Explanation of Figure 5(a)
>
> We apologize for the lack of clarity. In Figure 5(a), the X-axis represents the **model cost constraint** (specifically, the parameter count), and the Y-axis represents the **accuracy** achieved by the router under that constraint.
>
> A point $(X, Y)$ on a router's curve (e.g., K-NN) means, "When the router is restricted to selecting models with at most $X$ parameters, it achieves an overall test accuracy of $Y$."
> The curves thus illustrate the cost-performance trade-off. A superior router achieves higher accuracy ($Y$) for a given cost budget ($X$). The "Individual Models" (dots) and "Heuristic" curve provide baselines; the fact that learned routers (like K-NN) lie above these baselines demonstrates their ability to select better models than a static choice for a given budget. We will revise the figure caption and description to make this interpretation explicit.

---

> > ### Comment · Reviewer_3m3M · 2025-11-26
> >
> > Thank you for your responses. I believe the superiority of kNN overall parametric training methods is a very interesting topic to investigate for future work. I have no more questions.

---

### Official Review · Reviewer_9rhj · 2025-11-03

**Soundness:** 3
**Presentation:** 3
**Contribution:** 2
**Rating:** 4
**Confidence:** 4

**Summary:**

This paper studies the limitation of previous LLM routing benchmarks from three perspectives: (1) limited task diversity, (2) imbalanced model pools, and (3) oversimplified evaluation methodologies. Given that, this paper proposes a novel evaluation framework that incorporates diverse task distributions, a balanced model pool of 85 models with complementary model strengths, and multi-faceted metrics.

**Strengths:**

1. How to properly evaluate LLM routing approaches is an important topic in current research on LLM serving systems.
2. This paper proposes a novel benchmark with 33k queries across 68 categories, which serves as a concrete foundation for effective LLM evaluation.
3. This paper illustrates the technical developments with sufficient justification.

**Weaknesses:**

1. Some advanced routing work is neither compared nor discussed. For example,
    1. Ding, Dujian, et al. "BEST-Route: Adaptive LLM Routing with Test-Time Optimal Compute." Forty-second International Conference on Machine Learning.
2. This paper primarily leverages binary label & BARTscore as the quality metrics for LLM responses, which seems limited for open-ended conversation — the mainstream LLM service scenarios.
3. In the evaluation section, the performance results are primarily reported on Llama2 models (fig 3 & 6) which are often considered as outdated models given the presence of Llama3 herd and Llama4 family.

**Questions:**

1. Majority of the evaluation examples are just one-turn query. However, the daily LLM usage typically happens with multi-turn conversation. Can the proposed benchmark to evaluate LLM routing performance in the multi-turn conversation scenarios?

---

> ### Author Response · Authors · 2025-11-25
> **Thanks!**
>
> ## [W1] Including advanced routers – we expanded experiments, yet our focus is the evaluation methodology
>
> We thank the reviewer for this excellent suggestion. Our paper's primary focus is on **the evaluation methodology itself**, showing how benchmark design (task diversity, duplication, and model dominance) can bias router assessment. We chose to evaluate a representative set of classes of routers (clustering-based, learning-based) rather than to provide a comprehensive empirical survey.
>
> However, we agree that demonstrating our benchmark's utility with more advanced, state-of-the-art routers is valuable, we have now expanded our experiments to include state-of-the-art router algorithms: specifically, MIRT and NIRT routers [1]. Importantly, we evaluate both routers on the original EmbedLLM benchmark and on our remastered version. The table below summarizes our findings:
>
> *Table: Performance of MIRT and NIRT on the original vs. remastered benchmark of EmbedLLM.*
>
> | Benchmark   | MIRT AUC | MIRT Peak Acc | NIRT AUC | NIRT Peak Acc |
> | :---------- | -------: | ------------: | -------: | ------------: |
> | Original    |   53.46  |        64.27  |   50.06  |        58.00  |
> | Remastered  |   54.31  |        64.77  |   50.79  |        58.62  |
>
> These results demonstrate that: (1) our remastered benchmark can indeed surface subtle but meaningful differences in performance for advanced routers; and (2) key methodological issues we highlight persist even for these state-of-the-art approaches, supporting our central claim that evaluation quality determines the fairness and informativeness of routing results.
>
> We note that a truly comprehensive empirical benchmark of all advanced and emerging routers (e.g., including BEST-Route and others) would require substantial engineering and is beyond our current scope. However, our benchmark is modular and open-source, and we explicitly encourage the community to use it as a standard platform for future router comparisons—especially of advanced models.
>
> [1] IRT-Router: Effective and Interpretable Multi-LLM Routing via Item Response Theory. ACL 2025. arxiv.2506.01048.
>
> ## [W2] Regarding open-ended metrics – we prioritize reproducibility over coverage
>
> This is a valid and important limitation of all current large-scale, automated benchmarks, including our own. We chose binary labels (from EmbedLLM/RouterBench) and BARTScore (from MixInstruct) because they are **established, objective, and reproducible metrics in the literature**.
>
> - As we note in our own critique of MixInstruct (Appendix J), even metrics like BARTScore are flawed, as they "conflate... quality with similarity to GPT-4."
> - Designing a reliable, scalable metric for open-ended conversation is itself a major open research problem.
>
> In this work, we **deliberately prioritize reproducibility**, which we view as a cornerstone of a good benchmark. We added a discussion in the limitations section to make this trade-off (reproducibility vs. coverage of open-ended quality) more explicit.
>
> ## [W3] Llama 2 obsolescence – we follow other router works and the evaluation framework is model-agnostic
>
> Thank you for raising this important concern regarding Llama 2 model obsolescence. We acknowledge that the main results in our evaluation are based on Llama 2 models, which have now been surpassed by newer families such as Llama 3, Llama 4, and others. This choice was primarily due to **the availability of standardized benchmark datasets and open-source router implementations**, which are currently more mature and widely adopted for Llama 2.
>
> However, the **central contribution of our work is the development of an open-source, modular, and model-agnostic evaluation framework** for router assessment. Our framework is designed so that any new set of models, including Llama 3, Llama 4, or future architectures, can be easily integrated, with all benchmarking and metrics functioning identically. We make this extensibility explicit in the public codebase, and the framework’s model adapters support rapid onboarding of future LLMs with minimal modification.
>
> ## [Q1] Multi-turn support – future work based on single-turn foundation
>
> This is an excellent direction for future work. Our current benchmark (and the routers we test) focuses on **atomic, single-turn query routing**. Multi-turn conversation introduces the significant and distinct challenge of stateful routing (e.g., routing based on conversation history). The problems we identify and the metrics we define are conceptually compatible with multi-turn settings, but supporting this would require additional design choices for how routers consume and use dialogue history. We view this as an important and separate research problem that lies outside the scope of the present work.

---

### Official Review · Reviewer_gUhH · 2025-11-04

**Soundness:** 2
**Presentation:** 3
**Contribution:** 2
**Rating:** 4
**Confidence:** 4

**Summary:**

The paper addresses fundamental flaws in current benchmarks for evaluating systems that dynamically route queries to the most suitable large language model (LLM). Existing frameworks often suffer from limited task diversity, imbalanced model pools, and oversimplified metrics, leading to misleading conclusions about router performance. To overcome these issues, the authors propose RouterBench+, a comprehensive evaluation framework featuring (1) a diverse and realistic task distribution of 33,337 queries across 68 categories, (2) a balanced pool of 85 models with complementary strengths, and (3) modified metrics that better capture the cost-performance trade-offs and out-of-distribution robustness.

**Strengths:**

1. The paper accurately identifies important flaws in existing router evaluation pipelines. Lack of task diversity, dominance by a single model, and poor assessment of OOD performance are all major issues that make existing evaluation pipelines unsuitable for real-world scenarios.

2. The solutions are clearly and systematically presented, with experiments and plots illustrating their efficacy, and overall, the paper is easy to follow.

**Weaknesses:**

1. The removal of duplicate queries leads to very small improvements for learning based routers and causes a drop in performance in for clustering-based routers. Thus, the claim in lines 268-269 that "duplicate queries with conflicting labels can mislead routers" is not well substantiated by the results.

2. The role of the pseuod-specialist models in Section 4.3 is not clear. If they are not meant for deployment, how does adding them provide any benefit? If a router is selected based on its performance in the presence of pseuod-specialist models, but does not have access to those models when it is deployed, how will it be able to replicate that performance?

3. It is not clear how the binary routing evaluation paradigm in Fig 3 is helpful in evaluating routers in multi-LLM settings.

4. It is not clear how the benchmark incorporates cost-performance trade-offs, latency constraints and reliability, as claimed in line 420

**Questions:**

1. Please give some examples of common-sense and domain-specific tasks to better illustrate the difference between the two.

2. How is the generalist model and the set $\mathcal{M}_{\text{non-gen}} $ chosen from a given set of models?

3. The expression in line 237 suggests that the score is the difference between specialist and generalist accuracy but the caption of figure 2 says it is the difference between the specialist and the heuristic router. Which is it?

4. What is the dimensionality of the embeddings $\mathbf{e}_i$, $\mathbf{e}_j$ in line 263? A high dimensionality may lead to inconsistent results due to the curse of dimensionality.

5. How will sub-sampling tasks in EmbedLLM help (Section 5.1) if the dataset has very few specialist tasks to begin with?

---

> ### Author Response · Authors · 2025-11-24
> **Thanks! (1/2)**
>
> ## [W1] The impact of duplicate removal – we illustrate brittleness
>
> We thank the reviewer for raising this point. In this part of the paper, **our main goal is to draw the community's attention to the issue of duplicate queries with conflicting labels**, and we therefore conducted a small-scale experiment to illustrate what happens when we remove them. The duplicate problem clearly exists, but the current experimental numbers are not intended as definitive proof; rather, they can be interpreted as reflecting method-specific characteristics. Importantly, the changes in accuracy across routers substantiate our main contribution: **current evaluation protocols are brittle, since the reported performance can change when such artifacts are removed**.
>
> To further substantiate this, we observed that removing duplicates increases the **discriminative gap** (the performance margin between adaptive routers like k-NN and the static heuristic) from 3.69 to 3.95 AUC points in our remastered evaluation. While the absolute performance shift may appear small, this consistent widening of the gap confirms that duplicates were indeed acting as noise that compressed the performance distinction between methods.
>
> Our experiments indicate that (i) for clustering-based routers (K-NN/KMeans), performance drops when duplicates are removed, suggesting that these methods are highly sensitive to their immediate neighbors and were partially overfitting to noisy, conflicting artifacts in the original dataset; and (ii) for learning-based routers (MLP), performance improves after removing duplicates, suggesting that the label noise was previously misleading the model, and that cleaning it allows the router to learn a more generalizable decision boundary.
>
> We rewrote the paragraph (Lines 268–269) to state this conclusion explicitly and to clarify that these experiments are illustrative evidence of the problem rather than a formal proof.
>
> ## [W2] The role of pseudo-specialists – it is a diagnostic tool for adaptability
>
> We thank the reviewer for this concern and for allowing us to clarify the role of pseudo-specialist models.
>
> The goal is not to replicate specific performance metrics at deployment time. Instead, **pseudo-models are a deliberate methodological choice to create a diagnostic "stress test"**. In standard benchmarks, single-model dominance often compresses the performance gap between simple heuristics and sophisticated routers, making it difficult to assess a router's true decision-making capabilities. By surgically introducing pseudo-specialists, we create a **controlled environment** where model dominance is broken. This allows us to:
>
> 1. **Uncouple** the evaluation of the router's logic from the static quality of the model pool.
> 2. **Rigorously test adaptability**, verifying if a router can detect and leverage a specialist when one is actually available.
>
> Table 4 demonstrates the benefit of this design: adaptive routers (like K-NN, KMeans, MLP) successfully exploit these pseudo-specialists (e.g., a 31.03% reduction in agreement with the generalist for KMeans on LogiQA), while static heuristics do not. This proves that our framework can distinguish capable, adaptive routers from static baselines, whereas existing benchmarks largely cannot.
>
> ## [W3] Utility of binary routing – it is a complement to deferral curves
>
> We appreciate the reviewer's concern and would like to clarify the role of this component. **This is an important addition to the multi-LLM deferral-curve paradigm.** Specifically, the binary routing evaluation is designed to *complement* existing cost-evaluation paradigms. While deferral curves (used in RouterBench and in our work) are excellent for showing overall performance at a given cost budget, we introduce a "binary routing evaluation paradigm" (Figure 6) to provide a more direct view of cost-efficiency.
>
> This paradigm explicitly evaluates a router's ability to substitute a smaller/cheaper model for a larger one, and it maps out the full trade-off curve between accuracy and the rate of using the expensive model. We believe this provides a more fine-grained tool for analyzing a router's cost-saving behavior in multi-LLM settings.

---

> ### Author Response · Authors · 2025-11-24
> **Thanks! (2/2)**
>
> ## [W4] How cost, latency, and reliability are measured – they are mapped to parameters and OOD performance
>
> Thanks for pointing out this broad summary line. We will revise it to be more precise. The benchmark incorporates these aspects as follows:
>
> - Cost–Performance Trade-offs: This is the central function of our deferral curve (Fig. 5a), which explicitly plots Accuracy (performance) against Model Parameters (a proxy for cost).
> - Latency Constraints: We acknowledge in Appendix F that we use parameter count as a reproducible proxy for cost/latency, a common practice in the literature. However, our framework is explicitly extensible to real-world latency metrics, as detailed in Appendix F: "our framework supports per-query metrics (tokens, wall-clock latency)".
> - Reliability (Robustness): This is precisely what our OOD testing framework (Sec. 4.4) and the results in Table 5 are designed to measure.
>
> ## [Q1] Please give some examples of common-sense and domain-specific tasks to better illustrate the difference between the two.
>
> As stated in the paper, PIQA is an example of a common-sense task where generalist models perform well. It is a multiple-choice benchmark for physical commonsense reasoning: each question describes an everyday goal (e.g., “you want to apply eyeshadow but don’t have a brush”) and asks the model to choose the more plausible of two short candidate solutions (e.g., using a cotton swab vs. a toothpick). Similarly, TruthfulQA targets general world knowledge and commonsense misconceptions, asking models to answer questions such as “Can you get sunburn on a cloudy day?” in a way that reflects human commonsense. Such knowledge is broad, overlaps heavily with web text and everyday experience, and is therefore well covered by large generalist pretraining.
>
> In contrast, MedMCQA is a domain-specific medical multiple-choice benchmark constructed from real medical entrance exams. It contains long, technical questions that hinge on detailed clinical facts, such as drug mechanisms or specific lab abnormalities. Likewise, MMLU–Professional Law consists of bar-exam-style questions that require precise legal knowledge. These domain-knowledge-heavy questions are under-represented in generic web corpora, so generalist models tend to struggle, whereas specialist models further trained on curated medical or legal data are much better suited to this setting.
>
> ## [Q2] How is the generalist model and the set M_non-gen chosen from a given set of models?
>
> You are referring to the calculation of the specialist score. For a given collection of models, we first identify **a generalist model as the one with the highest overall average performance across all tasks**. The set of non-generalist models $M_{non-gen}$ (our hypothetical “specialists”) is simply **the rest of the models after excluding this best overall model**.
>
> Intuitively, a true specialist should excel in a particular domain and thus create a large positive accuracy gap over the generalist model on tasks from that domain, while possibly underperforming elsewhere. Our specialist score is designed to capture this pattern. However, in our experiments we rarely observe such large, domain-localized gains: most models that are supposedly domain-specialized do not substantially outperform the generalist model even on the tasks where they are expected to specialize.
>
>
> ## [Q3] The expression in line 237 suggests that the score is the difference between specialist and generalist accuracy, but the caption of figure 2 says it is the difference between the specialist and the heuristic router. Which is it?
>
> We apologize for the confusion. **The best-performing generalist is the heuristic model**, so the two descriptions are in fact referring to the same quantity in the specialist score calculation. We revised the text and caption to use consistent terminology.
>
> ## [Q4] What is the dimensionality of the embeddings e_i and e_j in line 263? A high dimensionality may lead to inconsistent results due to the curse of dimensionality.
>
> The dimensionality is 768. This choice means that, in principle, the "curse of dimensionality" could be a concern, and part of our goal is to highlight that such issues may affect router behavior. We set $\delta = 0.999$ to be very high so that the queries are essentially identical in embedding space, and we provide this as an illustrative example.
>
> ## [Q5] How will sub-sampling tasks in EmbedLLM help (Section 5.1) if the dataset has very few specialist tasks to begin with?
>
> We agree that this is an important concern, and **it is precisely what we hope to improve in evaluation**. If there are few specialist tasks, then **a comprehensive evaluation should incorporate more such tasks**, like in [1]. Our methodology is designed to make this deficiency visible and to guide the construction of better benchmarks and evaluations, rather than a new dataset.
>
> [1] RouterArena: An Open Platform for Comprehensive Comparison of LLM Routers

---

### Author Response · Authors · 2025-11-29
**Summary of rebuttal**

We would like to thank all reviewers for their constructive feedback and for recognizing the strengths of our work, specifically:

- **Important and Timely Problem:** Reviewers acknowledged the significance and timeliness of systematically analyzing LLM router evaluation, noting that current benchmarks suffer from critical flaws that need to be addressed. (Reviewers `gUhH`, `9rhj`, `3m3M`, `E1wn`)
- **Complete and Coherent Empirical Findings:** The empirical evaluation was recognized as thorough and well-structured, covering multiple benchmarks, router types, and evaluation scenarios. (Reviewers `3m3M`, `E1wn`)
- **Sound and Meaningful Solutions:** The proposed remastered evaluation pipeline—including de-duplication, pseudo-specialist injection, specialist score metric, and task diversity improvements—was recognized as methodologically sound and providing meaningful insights for router assessment. (Reviewers `gUhH`, `9rhj`, `E1wn`)
- **Clear and Accessible Writing:** All reviewers rated the presentation as good, finding the paper well-written and easy to follow. (Reviewers `gUhH`, `9rhj`, `3m3M`, `E1wn`)

During the rebuttal, we addressed several common concerns and provided additional experiments to strengthen our claims:

- **Role of pseudo-specialists:** Multiple reviewers asked about the utility of pseudo-specialist models. We clarified that these are **diagnostic tools**, not deployment artifacts—they create controlled "stress tests" to evaluate whether routers can detect and exploit specialists when available, decoupling routing logic from model pool quality.
- **Impact of our proposed fixes:** We demonstrated that our methodological improvements (de-duplication, specialist up-weighting) nearly **double the discriminative gap** between adaptive routers (e.g., k-NN) and static heuristics (3.69 → 6.00 AUC), making it easier to distinguish strong routing methods from weak ones.
- **Extended analysis on additional benchmarks:** In response to reviewer requests, we added analysis of the CARROT dataset and experiments with state-of-the-art routers (MIRT, NIRT), showing that the structural issues we identify persist across benchmarks and affect even advanced routing methods.
- **Model and embedding agnosticism:** We emphasized that our framework is designed to easily incorporate newer model families (Llama 3/4) and different embedding models, with the current focus on Llama 2 driven by data availability and reproducibility.

We appreciate the reviewers' engagement and believe it is crucial to bring these evaluation issues to the community's attention sooner rather than later, fostering discussion toward establishing good and reliable router evaluation practices that are currently missing in the field.

---

### Note · Authors · 2026-01-28

I have read and agree with the venue's withdrawal policy on behalf of myself and my co-authors.

---

### Meta-Review · Area_Chair_9X5p · 2026-01-01

**Summary:**

Strengths pointed out by reviewers:
- The work points out important flaws in existing router evaluations.
- Presentation is clear and systematically.
- Proposal of a new benchmark.
- Consideres OOD generalization.

Weaknesses mentioned by reviewers:
- The claim in lines 268-269 that "duplicate queries with conflicting labels can mislead routers" is not well substantiated by the results. **Addressed.**
- The role of pseudo-specialist models is not clear. **Addressed**
- Unclear how the binary routing evaluation is helpful. **Partially Addressed**.
- Unclear how the benchmark incorporates cost-performance trade-offs. **Addressed**.
- Relevant Related work missing.
- Binary label and BARTscores appear limited for open-ended conversations. **Partially addressed.**
- Results are primarily reported on the (by now) outdated LLama2. **Not addressed.** The evaluation was not extended to cover other model families.
- How does the embedding model influence the router? **Partially addressed.**
- Why do kNN-based methods outperform trained MLPs? Can you add a simple parametric baseline? **Addressed.**
- Expanding the evaluation: How are other predictive routing evaluations such as CARROT or router bench impacted by the identified flaws? **Addressed**
- The specialist models feel somewhat contrived. **Partially addressed**
- Current routing methods are not evaluated on this new benchmark. **Partially addressed.** It would have been good to see several state of the art routing methods and how they compare on the new benchmark. As such, it is unclear, how the routing rankings change.

Questions raised by the reviewers:
- Please give some examples of common-sense and domain-specific tasks to better illustrate the difference between the two. **Addressed**
- How is the generalist model and the set $\mathcal{M}_\text{non-gen}$ chosen from a given set of models? **Addressed**
- The expression in line 237 suggests that the score is the difference between specialist and generalist accuracy but the caption of figure 2 says it is the difference between the specialist and the heuristic router. Which is it? **Addressed**
- What is the dimensionality of the embeddings ,  in line 263? A high dimensionality may lead to inconsistent results due to the curse of dimensionality. **Partially addressed**. More explanation and insights around choosing $\delta$ would have been beneficial.
- How will sub-sampling tasks in EmbedLLM help (Section 5.1) if the dataset has very few specialist tasks to begin with? **Addressed.**
- Can the proposed router benchmark handle multi-turn scenarios? **Addressed.**

**Reviewer Concerns:**

See above.

**Reviewer Scores:**

- gUhH: $4 \to 6$
- 9rhj: $4 \to 4$: not all points where addressed. Particularly the evaluation on current models is missing.
- 3m3M: $6 \to 6$
- E1wn: $6 \to$ either $4$ or $6$. The reviewer pointed out that this is the most critical weakness. A comparison on several sota routing methods would have been beneficial here.

---

### Decision · Program_Chairs · 2026-01-26

Reject